# Benchmark Data Repositories for Better Benchmarking

**Rachel Longjohn**[1]*,    **Markelle Kelly**[2]*,    **Sameer Singh**[2],    **Padhraic Smyth**[2]

Department of Statistics[1], Department of Computer Science[2]
University of California, Irvine
Irvine, CA 92697
{rlongjoh,kmarke,sameer,pjsmyth}@uci.edu

## Abstract

In machine learning research, it is common to evaluate algorithms via their performance on standard benchmark datasets. While a growing body of work establishes guidelines for—and levies criticisms at—data and benchmarking practices in machine learning, comparatively less attention has been paid to the data repositories where these datasets are stored, documented, and shared. In this paper, we analyze the landscape of these *benchmark data repositories* and the role they can play in improving benchmarking. This role includes addressing issues with both datasets themselves (e.g., representational harms, construct validity) and the manner in which evaluation is carried out using such datasets (e.g., overemphasis on a few datasets and metrics, lack of reproducibility). To this end, we identify and discuss a set of considerations surrounding the design and use of benchmark data repositories, with a focus on improving benchmarking practices in machine learning.

## 1   Introduction

Evaluating machine learning (ML) algorithms on benchmark datasets is a central pillar of ML research. This performance benchmarking facilitates direct comparison across different techniques, which is important, for example, in the publication of research that introduces a novel method or for selecting the most appropriate approach for a particular application [1–5]. Ideally, these benchmark datasets serve as proxies for real-world tasks, so that performing well on the task represents meaningful advancement toward some desired real-world ML capability [6–10]. Benchmarking can help quantify progress on these tasks over time, and the availability of a well-studied, standard task evaluation environment can be a critical first step before moving to real-world applications, especially in high-stakes or expensive domains. In addition, evaluating with a benchmark dataset can be useful as a sanity check when developing a new methodology, as well as for ML education and training [11, 12].

Early data repositories, such as the UCI ML Repository, arose to address the data needs that come with ML benchmarking [13]. These repositories started as relatively small-scale efforts, but as the field of ML has rapidly grown, they have become more sophisticated, supporting additional features such as leaderboards comparing the benchmarked performance of ML models on a given dataset [14–20]. ML data repositories are fundamentally different from traditional domain-specific data repositories. For example, they tend to contain datasets from a wide variety of domains, and the process of selecting a dataset is often less about a scientific or engineering application and more about the compositional characteristics of the data and its associated tasks, for which a particular class of methods is applicable, e.g., multivariate spatiotemporal or network datasets.

---

*Denotes equal contribution.

Today, these ML data repositories—in particular, HuggingFace Datasets, Kaggle, OpenML, Papers with Code Datasets, TensorFlow Datasets, and the UCI ML Repository—are widely used. However, relatively little work has been devoted to understanding them and the specific factors involved in their design. In this paper, we introduce the term *benchmark data repository* to describe repositories that support the discovery and use of datasets for evaluating ML models (for brevity, we will use *benchmark repository* in the remainder of the paper). Our focus is the role of benchmark repositories in ML research, particularly the relationship between benchmark repositories and criticisms of current data and benchmarking practices in ML.[2] Each of Sections 2-6 reviews one of these issues, progressing through the dataset lifecycle—from creation and development to documentation and sharing to use and reuse for model evaluation [21–23]. For each criticism, we identify ways in which benchmark repositories can be part of the solution, motivated by existing standards, observed trends in the field, and examples from our experience as repository curators. We believe that these recommendations will be useful both for the owners of benchmark repositories in designing and improving their repositories, and more broadly to the creators and users of benchmark datasets in determining how to store, document, and find data. To the best of our knowledge, this paper is the first to define and establish best practices specifically for benchmark repositories, as well as to connect the practices of benchmark repositories to ML and data repository practices in general.

## 2 Valuing Datasets as Research Contributions

Data work maintains a legacy of being under-valued and under-incentivized by the ML community [6, 24–32], often regarded as an "engineering exercise" [33] or "operational" [26]. Several recent initiatives, such as the NeurIPS Datasets and Benchmarks track[3] and the Journal of Data-centric Machine Learning Research [34], have sought to change this pattern by providing peer-reviewed venues for publishing papers on data contributions. In this section, we posit that benchmark repositories can also help recognize datasets as intellectual contributions to the scholarly ecosystem by providing 1) dataset citations, 2) "connection metadata," and 3) dataset licenses. Reinforcing the value of data work incentivizes dataset creators to pay greater care and attention during dataset development and documentation—effects that propagate throughout the dataset lifecycle [24, 26, 27].

### 2.1 Dataset Citations and Metrics

For a dataset to operate in the ML ecosystem as a first-class research contribution, researchers must be able to locate it and its metadata via a persistent stable URL (as is the norm with published papers). In particular, the assignment of a *persistent identifier* (PID), such as a DOI, that can reliably be used to access a dataset has been widely recommended by experts [32, 35–39]. However, ML datasets and their documentation frequently lack PIDs and are often only available via GitHub or personal/research group websites [32, 40, 41]. Repositories can help address this by minting DOIs for submitted datasets (e.g., as Kaggle[4] and HuggingFace Datasets[5] do).

PIDs are the foundation of *dataset citations*, which give proper attribution to dataset creators, rather than solely citing associated publications [36, 42–49]. In ML, however, datasets are often referred to using combinations of names, descriptions, and associated papers, which can be challenging to disambiguate [40]. In contrast, many data repositories already provide standardized dataset citations that can be easily copied in a desired format (e.g., BibTeX) and include the minted DOI (Figure 1). Beyond giving credit, citing a dataset enables researchers to track its usage throughout the literature, which is particularly relevant in ML, e.g., for performance comparisons.

Furthermore, metrics such as the number of citations, number of views, or number of downloads can help quantify data impact, highlighting the value of the dataset in terms of its contribution to the ML community and potentially benefiting a variety of stakeholders (e.g., the researchers whose work is being cited or funders assessing a return on investment [50–53]). Repositories can provide the infrastructure for tracking these metrics of interest; for example, OpenML counts the number of

---

[2]While in this paper we focus on data used for model evaluation, we note that many of our points are also relevant to pretraining data.

[3]https://neuripsconf.medium.com/announcing-the-neurips-2021-datasets-and-benchmarks-track-644e27c1e66c

[4]https://www.kaggle.com/discussions/product-feedback/108594

[5]https://huggingface.co/blog/introducing-doi

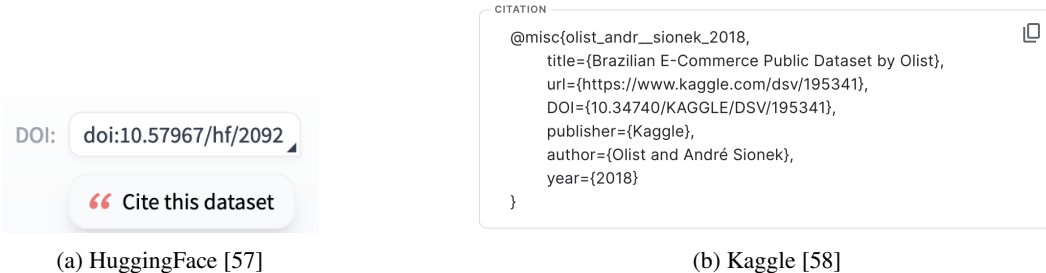

(a) HuggingFace [57]                    (b) Kaggle [58]

Figure 1: Examples of DOIs and citations in repositories.

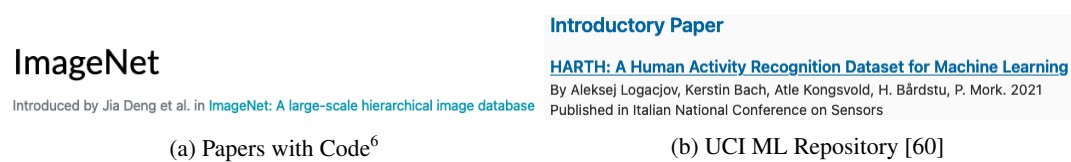

(a) Papers with Code[6]                    (b) UCI ML Repository [60]

Figure 2: Examples of connecting datasets to papers in repositories.

times a dataset has been used in experiment runs and the number of times it has been downloaded [14]. Data organizations, such as Scholix, the Data Usage Metric Working Group and Project Counter [54–56], are working towards more sophisticated frameworks for the provision of data metrics, and benchmark repositories are in a prime position to foster collaborations with these efforts.

## 2.2 Connection Metadata

Repositories can also support the treatment of datasets as research contributions via *connection metadata*, which connects a dataset to associated research entities (such as the dataset's creators or maintainers, publications, code, or other datasets) [59].

The dataset construction process is rife with consequential, "value-laden" decisions [8, 28, 29, 33]. The rationale behind these decisions may be described in an *introductory paper*: a publication that combines the narrative style of an article with the technical description of a dataset and its design process (also referred to as "data articles" [61] or "dataset descriptors"[7]). Introductory papers can give the data a story beyond standardized documentation, providing useful context about the problem, background on data collection procedures, and guidance about tasks for which the data have already been used. When these papers are peer-reviewed, it can lend additional credibility to the dataset for those considering it for re-use. Introductory papers can be included in benchmark repositories as a standardized metadata field (Figure 2), raising the visibility of these important documents.

Repositories can also identify an individual who agrees to serve as a dataset's *point of contact*: someone responsible for answering questions about the dataset and addressing any issues. Ideally, this person is also one of the dataset's creators, as they are best equipped to answer questions about the data and facilitate re-use [41, 62]. Although long-term data maintenance, and determining different stakeholders' responsibilities in that maintenance, remain challenging tasks [28, 63, 64], establishing a point of contact can help prevent the development of a disconnect between a dataset and its creators, which is not uncommon in ML [65–67]. By requiring that contact information for a responsible individual be specified in metadata, repositories can encourage an ongoing connection between the dataset, those who created it, and those who want to re-use it.

## 2.3 Dataset Licenses

It is widely recommended that datasets come with clear use guidance, often via a *license* [35, 39, 68]. These safeguards can help prevent the unintended use of data, an important part of respecting datasets as intellectual contributions. Data repositories can include licenses as part of a dataset's metadata. They can also make selecting a license easier on dataset donors, e.g., by showing which licenses

---

[6]https://paperswithcode.com/dataset/imagenet

[7]https://www.nature.com/sdata/journal-information

are popular, providing help text and links to the license language, or comparing the salient parts of different popular licenses. For ML datasets in particular, licensing can be complicated (e.g., it is ambiguous if models trained on a dataset count as "derivative work" [69]), and dataset licenses commonly used in other domains may not effectively restrict data use in ML (e.g., training commercial ML pipelines) [40, 70]. In addition, current licensing practices for ML datasets are often irregular, conflicting, poorly documented, or over-permissive given the dataset content [71]; in one survey of over 1800 text datasets, 69% had "unspecified" licenses on HuggingFace [72]. To help mitigate these issues, benchmark repositories can encourage the use of licenses that were constructed with ML use cases in mind, such as the Montreal Data License [69], or others, as more work is done in this area.

## 3 Addressing Issues with Dataset Content

Over the past decade or two, numerous issues with common benchmark datasets have been discovered, including technical flaws such as labeling errors and annotation artifacts [5, 73–75], privacy and copyright violations [40, 76–78], inclusions of hate speech or other harmful content [79, 80], representational biases [77, 81–83], and miscellaneous ethical issues [40, 66, 84]. Without clear documentation or careful data auditing, it is easy for these problems to go undiscovered well after a dataset's initial release and propagate harmful effects to downstream results [38, 78]. Further, even once an issue is discovered, updating or deprecating a dataset can be ineffective [40, 75]. Benchmark repositories can help detect and address dataset issues by collecting contextual metadata, performing quality reviews, and supporting the revision and deprecation of datasets.

### 3.1 Contextual Metadata

Benchmark datasets are often disseminated without detailed information about their broader context [9, 27, 28, 76, 85, 86]. By collecting *contextual metadata*, including information about a dataset's source, funding, collection, annotation, and preprocessing, benchmark repositories can illuminate the assumptions and motivations of dataset creators and flag potential dataset issues [6, 7, 30, 87–89]. To this end, several standards and schemata that include contextual metadata have been established, including Data Cards [90], datasheets for datasets [68], the Dataset Nutrition Label [91, 92], and the FAIR principles [35]. Such metadata can help ML practitioners detect issues earlier [93]; several "retrospective" datasheets for well-known datasets have demonstrated how contextual information raises red flags and could have contributed to earlier detection of data issues [77, 94].

In particular, information about the source of a dataset can alert data users to privacy or consent issues, representation biases, the potential for harmful content, or a mismatch with their target domain [95, 96]. For example, multiple facial image datasets include mugshots or surveillance camera footage [97–99]—raising red flags about the consent and privacy of the photographed individuals. Another example is the NIST Face Recognition Vendor Test dataset, which was funded by the U.S. Department of Homeland Security and contains data from the U.S. Mexican visa archive [100, 101]. In the use of this dataset for general facial recognition evaluation (e.g., [102]), its source and original intent are cause for concern about its transferability [76]. Generally, understanding the origins of a dataset can help ML researchers determine if it is appropriate for their use case, discouraging the use of benchmarks that are poor proxies for the task they are supposedly evaluating [6, 7, 23, 87, 95].

Data selection, filtering, and annotation processes are important design decisions that can significantly impact downstream performance [7, 38, 78, 103, 104]. One example is the systematic exclusion of text authored by or about marginalized groups in large, web-scraped text datasets due to curation and data filtering processes [29, 33, 38, 77]. Another pervasive issue is biased annotations, which are often crowd-sourced [27, 38, 105–108], e.g., as have been documented in the ImageNet dataset [7, 104]. However, these processes tend to be under-documented [29]; for instance, in a survey of over 100 papers introducing computer vision datasets, 36.6% did not provide any description of the human annotators; only 7.8% reported annotator demographics [32].

Clearly documenting data source, intent, collection, and processing procedures sheds light on these dataset issues early on in the data lifecycle. However, dataset creators do not necessarily prioritize metadata on their own [32, 41], and documentation is often scattered and unstandardized [76, 85, 93]. Benchmark repositories can work against this pattern by requiring dataset creators to provide detailed documentation of contextual information (and making it easily accessible), e.g., via an accompanying datasheet [68] and/or published introductory paper, which thoroughly describe a dataset's context.

| Housing type | Correct value | Value in repository |
|---|---|---|
| Own | 3 | 2 |
| Rent | 2 | 1 |
| Free | 1 | 3 |

Figure 3: The Statlog (German Credit Data) dataset [109], hosted by the UCI ML Repository, is a sample of customer records from a German bank, with the task of classifying each individual as a good or bad credit risk. In the repository documentation, 8 categorical variables have their levels mixed up or incorrectly described (e.g., see attribute 15, the type of housing the debtor lives in, above). Groemping [67] tracked down papers which describe the dataset's origins [110–113] to construct a proper code table. She donated the corrected dataset as the South German Credit dataset in 2019 [67] but the original dataset from 1994 has nonetheless been widely used in ML research.

## 3.2 Quality Review

Ideally, quality issues with benchmark datasets and their metadata are detected early and corrected; otherwise, these concerns should either be documented or used as a rationale to withdraw the dataset. For example, datasets containing personally identifiable information should not be released [27] and documentation errors (as in Figure 3) should be quickly amended. Benchmark repositories can help identify these problems throughout the data lifecycle by (1) performing a pre-release *quality review* [95] to catch issues before a dataset is shared, and (2) by serving as a centralized location to collect users' reports and concerns [75] to flag issues throughout a dataset's use and reuse. With stringent quality assurance, ML researchers can reliably look to a repository for high-quality datasets [38], making it easier to avoid using unvalidated, problematic datasets for benchmarking.

Quality reviews can help counter the current lack of incentive for ML dataset creators to consider ethical issues, which has been pointed to as a major contributor to the numerous ethical problems with benchmark datasets [27]. For example, benchmark data collection often does not undergo institutional ethical review [78]; in one survey [32], only 5 out of 100 papers introducing datasets with human subjects mentioned an institutional review board (IRB) or equivalent ethical review. As a result there has been a call for more intervention in data curation, involving curators who can focus on developing conduct codes and ethical review processes rather than relying on dataset creators [27, 30, 33, 88]. It is an open question to what extent repositories should be involved in these decisions; several popular repositories (e.g., Zenodo, Mendeley) view their role as only providing infrastructure and not conducting any kind of data review. However, we posit that benchmark repositories are well-positioned in the data pipeline to perform at least basic ethical checks and initiate a movement towards interventionism. We point to the growing body of literature on ethical data curation for ML [30, 78, 89, 95, 106, 114] as a starting point for the development of ethical review processes.

Conducting thorough quality assurance can be particularly difficult for benchmark repositories because they typically host data from a variety of domains. In contrast, disciplinary repositories, which specialize in a particular domain, often have a community of experts with the knowledge to conduct quality reviews. We point to requiring peer-reviewed introductory papers as a potential step in this direction, as the publishing venue may be able to perform more targeted reviews, and an increasing number of venues also incorporate ethical reviews.[8] Repositories could also outsource reviews for datasets via a network of experts such as the Data Curation Network.[9]

## 3.3 Dataset Revision and Deprecation

Although quality review can help catch serious issues before the release of a dataset, inevitably, some datasets will need to be updated, corrected, or deprecated. As a centralized data source, benchmark repositories can help support the revision and deprecation of datasets.

Benchmark repositories can support *dataset revision* by documenting data versions and connecting each dataset to a responsible point of contact. When different versions of a dataset are not clearly associated with unique version numbers, differing versions may be used interchangeably [117, 118] (e.g., as in Figure 4). Repositories can enforce versioning by assigning a new version number whenever a data file is changed. Documentation of the revision, including what was changed or

---

[8]e.g., https://medium.com/@icml2024pc/ethics-review-at-icml-e3b4ce1afd54,https://neurips.cc/public/EthicsGuidelines
[9]https://datacurationnetwork.org

| Index | Sepal length | Sepal width | Petal length | Petal width |
|---|---|---|---|---|
| 35 | 4.9 | 3.1 | 1.5 | 0.2 |
| 38 | 4.9 | 3.6 | 1.4 | 0.1 |
| 90 | 5.5 | 2.5 | 4 | 1.3 |

(a) The original data for the 35th, 38th, and 90th iris flowers.

| Index | Sepal length | Sepal width | Petal length | Petal width |
|---|---|---|---|---|
| 35 | 4.9 | 3.1 | 1.5 | 0.2 |
| 38 | 4.9 | 3.6 | 1.4 | 0.1 |
| 90 | 5.5 | 2.5 | 5 | 1.3 |

(b) An alternative version of the data with the petal length of the 90th flower incorrect.

| Index | Sepal length | Sepal width | Petal length | Petal width |
|---|---|---|---|---|
| 35 | 4.9 | 3.1 | 1.5 | 0.1 |
| 38 | 4.9 | 3.1 | 1.5 | 0.1 |
| 90 | 5.5 | 2.5 | 4 | 1.3 |

(c) An alternative version of the data with erroneous entries for the 35th and 38th flowers.

Figure 4: The Iris dataset from the UCI ML Repository is widely used for evaluating clustering and classification algorithms [115]. Each observation corresponds to an iris flower, including sepal and petal measurements and its specific species (out of three classes). After years of use, it was discovered that there were multiple different widely-publicized versions of this dataset, with differing measurements for certain observations. Consequently, the reported performances of classification models on Iris (across a large number of published papers) are not necessarily comparable [116].

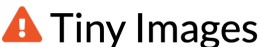 Tiny Images

Introduced by Antonio Torralba et al. in 80 Million Tiny Images: A Large Data Set for Nonparametric Object and Scer

⚠ This dataset has been retracted and should not be used

The image dataset TinyImages contains 80 million images of size 32×32 collected from the Internet, crawling the words in WordNet.

**The authors have decided to withdraw it because it contains offensive content, and have asked the community to stop using it.**

Figure 5: The Papers with Code dataset page for the deprecated Tiny Images dataset.

removed and a rationale for the changes, should also be provided [32, 95]. In addition, by associating datasets with a responsible point of contact (see Section 2.2), repositories can help streamline the resolution of questions or issues regarding a dataset.

Repositories can also support the *deprecation* of datasets. Currently, there is no standardized process for dataset deprecation: creators often withdraw their dataset without an explanation of why it was withdrawn or explicit instructions not to use the dataset (or other post-deprecation protocols). For example, in [75]'s case study of six high-profile dataset retractions, three (MS-Celeb-1M, Duke MTMC, and HRT Transgender) did not provide any reason for the dataset's removal. Moreover, deprecation reports are posted in a scattered, decentralized manner via news articles, conference papers, or researcher or lab websites [75]. Ultimately, it can be unclear to researchers if a dataset is acceptable to use; it is not uncommon for datasets to remain in use after their deprecation, including in published, peer-reviewed papers [7, 40, 75]. To mitigate this, benchmark repositories can (1) establish a process for deprecating a dataset, in which its creators submit a standardized report, detailing the reasons for deprecation and post-deprecation protocols, and (2) maintain a page connected to the dataset DOI (see Section 2.1) with the deprecation report and original metadata [119]. If a deprecation report is clearly displayed in the same place where a dataset was available, it clarifies to researchers (and reviewers) that the dataset should not be used (e.g., see Figure 5).

## 4  Promoting Data Usability and Reproducibility

Recent work has pointed to a need for improved (re)usability of data [28, 122] and reproducibility of benchmarked results [123–127] in ML. When a dataset lacks clear metadata, it can lead to critical misunderstandings in the reuse phase of its lifecycle (as in Figure 6). Unambiguous metadata are also a necessary foundation for benchmark reproducibility, ensuring that data are used in the same way across evaluations. Benchmark reproducibility is critical for ML research: it enables the verification of published results, provides a starting point for experimentation and follow-up work, and makes contributions easier for others to use, potentially increasing research impact [5, 128, 129]. Benchmark repositories can support usability and reproducibility with the metadata they require and provide to

| Variable name | Meaning |
|---|---|
| mcv | Mean corpuscular volume |
| alkphos | Alkaline phosphotase |
| sgpt | Alamine aminotransferase |
| sgot | Aspartae aminotransferase |
| gammagt | Gamma-glutamyl transpeptidase |
| drinks | Number of half-pint equivalents of alcoholic beverages drunk per day |
| selector | Field used to split data into train/test sets |

Five input features from blood tests

Actual target variable; misinterpreted as 6th feature

Misinterpreted as a binary target variable

Figure 6: The BUPA Liver Disorders dataset is a popular classification benchmark from the UCI ML Repository [120]. Each row contains information on an individual's consumption of alcoholic drinks and their results on several blood tests targeting alcohol-related liver issues; the intended task is to predict alcohol consumption based on these test results. The last column of the dataset is an indicator, added by the dataset creators, intended to split the rows into training and test sets; however, the data documentation did not clearly explain the meaning of each column. It was subsequently found that many highly-cited papers using this dataset had mistakenly treated this last column as the class label, producing "meaningless results" [121].

users [130, 131], particularly compositional and task-specific metadata (in addition to responsible points of contact and dataset versions—see Sections 2.2 and 3.3 [123]).

If repositories include benchmarked results alongside datasets (discussed further in Section 5), they can further support reproducibility by collecting and providing metadata about the benchmarked results themselves [123].

## 4.1   Compositional and Task Metadata

Datasets are more usable for ML researchers when accompanied by metadata describing the dataset composition and relevant tasks [35, 39, 132–134]. These metadata help expedite onboarding for a new dataset [41, 135], prevent misunderstandings, and promote data reproducibility [7, 136–139].

*Compositional metadata* describe the makeup of a dataset, e.g., for tabular datasets, what each instance represents, descriptions of each feature, the total number of rows and columns, the dataset label or target, the presence of missing data, and recommended data splits [68, 140]. When this information is clear, datasets are more interoperable, meaning they can be more easily processed and incorporated into different workflows [35, 135]. For example, a researcher might want to evaluate an existing model on a new dataset; if this dataset has detailed compositional metadata, the researcher can quickly and easily determine which columns they need to use and what preprocessing is required. If this metadata follows a standardized schema (e.g., Croissant [122]), model evaluation may even be done automatically or semi-automatically.

*Task metadata* include the intended or appropriate ML tasks for a dataset (e.g., image classification or time-series prediction) and specialized metadata relevant to those tasks (e.g., for human [78] or medical [136] image, NLP [141], or ecological [139] data). As an example, HuggingFace Datasets [17], which specializes in NLP data, collects metadata on language and multilinguality, text creation, and fine-grained NLP tasks (e.g., sentiment classification, multiple-choice question answering, or word sense disambiguation). Such specialized metadata make it easy for ML practitioners to find and use data that fit a specific application.

Thus, to improve data usability and benchmark reproducibility, repositories can require that data donors provide high-quality compositional and task metadata, including specialized task metadata, where appropriate [132]. Repositories can also help streamline metadata creation processes, which donors can find overwhelming or time-consuming [41, 142], e.g., with adaptive metadata collection, auto-filling or updating basic fields [41], or supplementary training and tools [143]. We note that to some extent, the responsibility to provide accurate and complete metadata ultimately falls on the data donor. However, repositories can reduce the risk of incorrect, incomplete, or manipulated documentation by enforcing metadata schemata, using quality review processes, and providing user-friendly metadata creation tools—taking some of this burden from the donors themselves.

## 4.2 Benchmark Metadata

In ML benchmarking, implementation details such as software dependencies, random seeds, and hyperparameter values can have a significant impact on results [5, 124, 144], as can the details of metric computation, including data splits, metric definitions, and aggregation of results [5, 15]. Thus, clearly documenting this *benchmark metadata* is critical for reproducibility [123, 145]. Beyond validating results, the ability to replicate an analysis facilitates "hands-on" experimentation with benchmarked models on a given dataset, enabling researchers to test potential modifications, perform additional evaluations, or debug other models [128, 146]. To this end, if a repository displays a particular benchmarked result for a dataset, they should ensure that specific details on all settings and hyperparameter values used to obtain that result are available [123]. The software environments, dependencies, code, and data files necessary to re-run analyses should also be documented and accessible [41, 117, 128, 147–149]. By ensuring that detailed metadata on a dataset's content, composition, task, and benchmarked results are available, repositories can provide ML practitioners with a holistic understanding of the benchmark [41, 150].

# 5 Encouraging Holistic Evaluation

It has become common practice to tabulate benchmarked metrics for different ML methods with a dataset leaderboard; several benchmark repositories offer leaderboard features, including Kaggle [18] and Papers with Code.[10] Leaderboarding has become predominant in ML evaluation, and state-of-the-art performance is a key factor in peer review processes [5, 151]. However, this leaderboard culture has been criticized for a "near singular" focus on the incremental improvement of a narrow set of metrics (e.g., classification accuracy) [7, 152]. Such fixation on a specific metric is unlikely to yield broadly applicable results and can stifle the growth of new, diverse ideas [5–7, 151]. Further, as measures of uncertainty are seldom incorporated, seemingly record-breaking performance improvements are not always statistically significant [151, 153, 154] (e.g., a review of the MS MARCO leaderboard found no significant difference in performance between the top three models [155]). In this paper, we refrain from taking a stance on whether benchmark repositories should include leaderboards. However, as several repositories currently act as centralized purveyors of leaderboards, we briefly discuss how they can promote more comprehensive evaluation, helping address problems with benchmarking that manifest later in the dataset lifecycle.

## 5.1 Analysis Beyond Single Metrics

Recent work on best practices for ML benchmarking recommends evaluating performance more holistically [6, 7, 156, 157]. This could include a variety of metrics, capturing model size and complexity, energy consumption, inference latency, and the amount of data used [1, 3, 5, 152, 158, 159]. Additional in-depth assessment—such as error analysis or disaggregated evaluations—can provide a more nuanced portrait of model behavior, capturing bias, fairness, or robustness [6–8, 106, 160]. Thus, to enhance their leaderboards, benchmark repositories can include this sort of comprehensive information on model performance. In addition to incentivizing progress in a number of dimensions, this approach reflects that the ideal model is context-dependent, enabling practitioners to choose a model based on the criteria most relevant to their use case [5, 7, 8, 158].

## 5.2 Metric Uncertainty

Metrics shown without any measure of uncertainty can prompt fallacious conclusions, e.g., that one model performs definitively better than another. Instead, including uncertainty makes these benchmarked results more informative [1, 161], and a growing body of methodologies has developed for estimating uncertainty, computing confidence intervals, and performing statistical significance testing in the context of model comparison [5, 144, 155, 162–164]. In light of this, several guidelines for ML evaluation call for the inclusion of variance, uncertainty, and statistical significance in model analysis [87, 95, 106, 151, 165]. Repositories with leaderboards can support this movement by enforcing the reporting of uncertainty alongside point estimates of metrics.

---

[10]e.g., https://paperswithcode.com/sota/image-classification-on-cifar-10

# 6 Diversifying Benchmark Datasets

Benchmarking for a particular type of ML model is often concentrated on a limited set of datasets and tasks [6, 8, 23]. This lack of diversity can encourage overfitting to a specific benchmark dataset (e.g., via random seed or hyperparameter fishing) [165, 166]. Over-adaptation also happens at a macro level over time, as new models leverage tricks and strategies from earlier work [5, 87]. Moreover, these benchmarks are often not directly relevant to the real-world behavior they are evaluating—for example, the GLUE benchmark [167] has been commonly used to evaluate natural language understanding, but it is mainly comprised of sequence matching tasks [5]. As a result, there is often a disconnect between benchmarked performance and real-world model behavior [10, 87, 168]. Thus, the overrepresentation of specific tasks, data types, and test datasets can ultimately bias long-term research directions [5, 6, 8] and limit the generalizability of model evaluations [9, 95, 106, 169]. To fight these patterns of overfitting and overuse, repositories can support the discovery and use of diverse, relevant, and continuously evolving datasets.

## 6.1 Living Datasets

To hinder the overfitting of models to a specific test set, leaderboards can evaluate submitted models on a private, hitherto unused test set [170–172] (e.g., as done by Kaggle) or on out-of-distribution data [151, 173]. Extending this principle, leaderboards can also support "living" or evolving datasets, to which dataset creators continuously add new examples or tasks and remove outdated or erroneous examples. While this means that the benchmarked performances of two models evaluated at two different points in time may not be directly comparable (and data versioning, as discussed in Section 3.3, is critical), robust models will generally outperform those using a specific trick or artifact as evaluations are repeated over time [5]. These living datasets also track the real-world evolution of data—for instance, in the context of autonomous driving, new types of vehicles appear on the roads [38]—which static benchmark datasets fail to capture [75]. By evaluating models on living datasets, repositories can help shift focus away from a specific static set of examples, de-incentivizing overfitting and helping bridge the gap between benchmarked and real-world performance.

## 6.2 Dataset Discoverability

When choosing benchmark datasets for evaluating an algorithm, it has become the default to select the same datasets already used in the literature [9, 76]. Often, however, there also exists a plethora of other high-quality datasets that could have been used but did not win the "benchmark lottery" [5] and were left undiscovered. To support *dataset discoverability*, existing standards emphasize the importance of standardized, rich metadata [39, 132, 133], which enable searching for datasets via keywords, filtering, and controlled vocabularies [174].

For benchmark repositories, this search is often task-driven: ML practitioners need to find datasets for which a certain type of model is applicable, based on compositional properties and relevant tasks (see Section 4.1). Thus, to improve benchmark dataset discovery, repositories can support search based on compositional and task metadata [33, 41, 130]—for example, the UCI ML Repository's search functionality includes a filter for classification, regression, clustering, or other datasets. Overall, by promoting the discovery and use of a more diverse set of evaluation datasets, repositories can build a barrier to the over-representation of specific benchmark tasks or datasets and encourage more generalizable model evaluation.

# 7 Discussion and Conclusion

## 7.1 Key Takeaways

A common thread throughout the criticisms of ML data and benchmarking practices we discuss in this paper is a need for the intervention of a third party—separate from dataset creators and users—in addressing these issues [5–7, 26–28, 31–33, 41, 75, 78, 95, 132]. While improving the state of ML evaluation will be a community effort, involving the efforts of conferences and journals, policymakers, nonprofit organizations, and individual practitioners [27], in this paper we posit that benchmark repositories can play a major role in this effort, instigating far-reaching changes to the culture surrounding datasets and benchmarking in ML. We summarize our key takeaways as follows.

- Repositories can highlight the status of datasets as valuable scholarly contributions.

- Repositories are well-positioned in the data pipeline to address issues with dataset content.

- Repositories can facilitate data reuse and benchmark reproducibility by ensuring salient metadata is provided for datasets (and, if applicable, benchmark evaluations).

- Repositories with leaderboard features can enforce best practices for model evaluation.

- Repositories can provide a platform for discovering new, relevant, high-quality datasets, counteracting the overuse of a small set of standard benchmark datasets.

## 7.2 Limitations

The long-term feasibility and impact of our suggestions are predicated upon larger shifts in community norms and attitudes about data-centric work, which will rely upon proper incentivization, an open challenge in the ML community [24, 27, 28]. We discuss here potential incentives for both individual researchers and repositories—although incentives for other actors (e.g., universities, companies, publishers) are also worth exploring.

A key incentive for researchers to become involved in repository efforts is funding; this is becoming more available as agencies such as the U.S. National Institutes of Health (NIH) and National Science Foundation (NSF) pay increasing attention to the data-sharing ecosystem.[11] For example, the NSF's program for Community Infrastructure for Research in Computer and Information Science and Engineering explicitly calls out funding support for data repositories,[12] and the U.S. National Artificial Intelligence Research Resource Task Force identifies repositories as important to their goal of "Strengthening and Democratizing the U.S. Artificial Intelligence Innovation Ecosystem" [135].

To incentivize dataset reviewers, repositories could follow the model of "volunteer journals" such as the *Journal of Machine Learning Research*. These public journals demonstrate how high-quality shared resources can be developed through dedicated volunteer efforts, offering inspiration for a parallel system of oversight, reviewing, and maintenance for repositories. For example, similar to the role of Action or Associate Editors (AEs) in these journals, repositories could have a set of curators who are responsible for identifying relevant experts to review a dataset. By framing data curation and review as an academic service in the same vein as more traditional editorial roles, repositories could help incentivize participation in the review process.

Further work is also needed to determine how to incentivize repositories themselves to enforce best practices (e.g., requiring data donors to select a license or provide task metadata). One potential avenue is to establish standards for benchmark repositories, building upon standards for data repositories in general, such as CoreTrustSeal.[13] Establishing standards or repository certification processes will be most effective if the ML community cultivates an expectation that such requirements are met (e.g., as in archival settings [27]).

Though we may draw useful inspiration from these ideas, it remains unclear what incentivization strategies will work best to spur large-scale buy-in from repositories, dataset creators, and dataset users in implementing and maintaining best practices.

## 7.3 Looking Ahead

Going forward, we hope that a growing appreciation of data work will permeate the ML community, serving as a catalyst for investment into data infrastructure in ML and broader researcher involvement in data repositories. In light of this, we believe the ideas in this paper lay a foundation for further discussion and research about how benchmark repositories can be utilized, and improved, for better benchmarking in ML.

---

[11]https://grants.nih.gov/grants/guide/notice-files/NOT-OD-21-013.html
[12]https://new.nsf.gov/funding/opportunities/circ-community-infrastructure-research-computer-information/nsf23-589/solicitation
[13]https://www.coretrustseal.org/

## Acknowledgements

This research was supported in part by the National Science Foundation under award CCRI-1925741 and in part by the Hasso Plattner Institute (HPI) Research Center in Machine Learning and Data Science at the University of California, Irvine.

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
