# OpenReview forum: "Benchmark Data Repositories for Better Benchmarking"
_NeurIPS.cc/2024/Datasets_and_Benchmarks_Track — NeurIPS 2024 Track Datasets and Benchmarks Poster_

### Official Review · Reviewer_axrk · 2024-06-18
**Good Paper Covering an Important but Underdiscussed Topic**

**Rating:** 8
**Confidence:** 4
**Clarity:** The paper flows well and is quite wel…

**Review:**

Pros:
- This paper provides an excellent overview of the current role of existing benchmarking repositories and the features they adopt to enable better dataset usage and construction practices.
- The recommendations put forth by the authors are both useful and also largely feasible. Some actions towards these points are taken by existing benchmark repositories, but the work is not complete.
- A greater awareness of this topic is important and will serve the community well.

Cons:
- The paper could benefit from a better exposition of how their desired norms can be enforced-as they point out, there are numerous examples of where datasets in a benchmark repository were still improperly used, and only discovered so after significant implications for research works.

Overall, the paper discusses an important topic and I believe would strongly benefit the community to learn from. I believe it may spur further work, either on the part of benchmark repositories to further improve, or on the part of other researchers.

**Strengths:**

This paper surfaces an important yet underdiscussed issue, presenting effective context for community members or newcomers who may not be especially familiar with what benchmark repositories are or their importance. The potential positive benefits of the best practices recommended by the authors are by and large quite high, toward fostering better benchmarking practices, healthier research norms, and better reproducibility.

**Additional Feedback:**

N/A.

**Correctness:**

To my knowledge, the paper's claims are correct. They appropriately surface existing work done by benchmark repositories when such exists.

**Documentation:**

N/A, the paper does not introduce new datasets.

**Ethics:**

No, no ethical concerns.

**Limitations:**

As the authors address, one limitation is that the recommendations require significant resources and buy-in on the part of benchmark repository creators and on the part of the community to create such new features, explain their usage adequately, and encourage change among the research community to adopt these better habits. However, this paper does make an effective first step at starting this conversation.

**Opportunities For Improvement:**

The authors might be able to propose steps that researchers who are not the maintainers of benchmark repositories can take to both encourage further development and further raise awareness. Such recommendations are made with respect to best practices for researchers alone within the paper, but discussing how benchmark repositories themselves can be benefited by the actions of researchers could provide further useful insights and advice.

**Relation To Prior Work:**

Yes, the paper adequately cites related works.

**Summary And Contributions:**

This paper provides an overview of "benchmark repositories", challenges they face, and provides recommendations for interventions benchmark repository maintainers can take to exert a positive impact on the field.

---

> ### Author Rebuttal · Authors · 2024-08-17
>
> Thank you for your detailed review and constructive feedback. We are very pleased that you found our paper to be a useful and high-impact contribution to the field and respond to your points in detail below.
>
> __“As the authors address, one limitation is that the recommendations require significant resources and buy-in on the part of benchmark repository creators and on the part of the community to create such new features, explain their usage adequately, and encourage change among the research community to adopt these better habits. However, this paper does make an effective first step at starting this conversation.”__
>
> Thank you for raising this important point, and we appreciate that you agree that our paper makes an effective first step in starting a discussion of these issues. To expand our discussion on the issue of incentivization in particular, we have added the following text to the Discussion section of the paper:
>
> _The long-term feasibility and impact of our suggestions are predicated upon larger shifts in community norms and attitudes about data-centric work, which will rely upon proper incentivization, an important open challenge in the ML community (e.g., refs 24, 27, 28). Incentivization is a topic that has attracted significant attention over the past decade by researchers who study ecosystems for sharing of research data (e.g., https://doi.org/10.1002/asi.22634, https://doi.org/10.5334/dsj-2017-014, https://mitpress.mit.edu/9780262529914/big-data-little-data-no-data/). Many of the lessons from this body of work are relevant in the context of research data repositories as well. We focus here on several potential incentives for individual researchers (which we note may differ from those for other actors e.g., universities, companies, or publishers), beyond the obvious incentives of contributing to the broad agenda of one’s research community and to the public good more broadly._
>
> _First, funding agencies such as the US NIH and NSF are paying increasing attention to the data-sharing ecosystem (https://grants.nih.gov/grants/guide/notice-files/NOT-OD-21-013.html). For example, the NSF’s program for Community Infrastructure for Research in Computer and Information Science and Engineering explicitly calls out funding support for data repositories, and the U.S. National Artificial Intelligence Research Resource Task Force identifies repositories as important to the success of their goal of “Strengthening and Democratizing the U.S. Artificial Intelligence Innovation Ecosystem” [130]. A potentially useful model for repositories here are “volunteer journals”, such as JMLR, which demonstrate how a community can develop high-quality shared resources through dedicated volunteer efforts. These public journals may provide lessons in how a parallel system of oversight, reviewing, and maintenance could be cultivated for repositories. For example, repositories could have a set of curators who are responsible for identifying relevant experts, much like how papers are reviewed for journals in which Action or Associate Editors call on specific experts to be reviewers. In this way, AEs and reviewers could get credit for this work, e.g., by adding to their CVs that they are an AE/reviewer for a repository or being publicly acknowledged by the repository. When considering promotions for researchers, universities are also becoming increasingly willing to recognize non-traditional service contributions, such as curatorial service related to research data repositories, even though traditionally such service has been in the form of editorial roles (e.g., for journals)._
>
> _Though we may draw useful inspiration from these ideas, it remains unclear what kind of incentivization will work best to spur large-scale buy-in from repositories, dataset creators, and dataset users into implementing and maintaining our recommendations._

---

> > ### Author Rebuttal · Authors · 2024-08-17
> >
> > __“The paper could benefit from a better exposition of how their desired norms can be enforced-as they point out, there are numerous examples of where datasets in a benchmark repository were still improperly used, and only discovered so after significant implications for research works.”__
> >
> > For many of our suggestions, it will be up to the repository to enforce the desired norms on data donors (e.g., requiring a license to be selected or task metadata provided). The larger question of how to enforce that _repositories_ implement best practices is also difficult and fits in well with the Limitations section we are adding to the discussion. The case studies of older datasets that we present show what can go wrong when repositories and users do _not_ adhere to best practices. There exist repository standards for data repositories in general (e.g., CoreTrustSeal https://www.coretrustseal.org/), so perhaps one avenue of enforcement is to implement a standard for benchmark repositories and try to build an expectation from the community that such a standard is met (e.g., as in archival settings https://doi.org/10.1145/3351095.3372829).
> >
> > __“The authors might be able to propose steps that researchers who are not the maintainers of benchmark repositories can take to both encourage further development and further raise awareness. Such recommendations are made with respect to best practices for researchers alone within the paper, but discussing how benchmark repositories themselves can be benefited by the actions of researchers could provide further useful insights and advice.”__
> >
> > This is a great point. While our paper indeed focuses on how benchmark repositories can help the field, benchmark repositories can also certainly benefit from researchers in turn. We see three potential benefits researchers can have on benchmark repositories:
> >
> > 1. We express in the paper that researchers/users should use benchmark repositories to store, document, and find data. Having more researchers using and engaging with repositories could promote larger cultural shifts in how the community regards data work, potentially leading to resource benefits for repositories, e.g., in funding and human labor, especially if a repository is able to build a reputation for having high-quality and useful datasets.
> > 2. Researchers can also be useful to benchmark repositories via dataset auditing. This is demonstrated via the case studies we share in the paper; researchers external to the repository took the time to audit older datasets, discovered important issues, and alerted the repository to help prevent further misuse.
> > 3. Having more researchers involved in repositories can also help build a community of reviewers. As we convey in the paper, how to conduct dataset reviews for benchmark repositories is still an important open question, but it would certainly help benchmark repositories to have a strong, diverse network of experts that they can contact for dataset reviews, e.g., having a neuroscientist open to reviewing a neuroscience dataset, or an NLP researcher to look at text datasets.
> >
> > We have added a discussion of these ideas for how researchers can benefit repositories to the last section of our paper; thank you for raising this point.

---

> > > ### Comment · Reviewer_axrk · 2024-08-26
> > > **Response to Authors**
> > >
> > > Thank you very much to the authors for their responses.
> > >
> > > The proposed additions are excellent and I believe they will further improve the paper. I believe this work will have a high positive impact on the NeurIPS D&B community.

---

### Official Review · Reviewer_iNCk · 2024-07-22

**Rating:** 6
**Confidence:** 4
**Correctness:** yes
**Clarity:** yes

**Review:**

The paper is well written and easy to read. It describes several of the important considerations regarding the curation of datasets in repositories and how certain characteristics of those repos like licensing, unique identifiers, and levels of metadata make them more future-proof, traceable and robust. Along the way they link into examples from literature of challenges and pitfalls that motivate their advocacy.

**Pros:** The issues the authors address re: benchmarking repositories are significant and they apply to a broad ML audience, many of whom use or curate these repositories.

**Cons:** The authors don’t explain clearly whether the recommendations they are making are a novel contribution in an of themselves. Many of the points they make and discuss are made elsewhere in literature, often in the papers introducing dataset repositories that they discuss. So it’s unclear how much of a novel contribution this paper provides.

**Strengths:**

The authors are addressing a significant and very relevant problem at a high level, one centered by this track at NeurIPS. The writing is clear and the recommendations are well-supported in literature throughout. The paper and recommendations are ethically well-aligned. I liked that the authors point out the misalignment of academic incentives with the work required to create benchmark repositories.

**Additional Feedback:**

I would have liked to see more insight into how _maintenance_ of benchmarking repositories could be made to align with academic incentives. The authors point out the issue that teams of people are needed to maintain benchmark repositories, and point to new venues of publication, but stop short of recommending any solutions to the problem that keeping benchmark resources up to date is not incentivized in the academic world that supports ML research - that is, once a paper on a resource is written, there is diminishing return in maintaining its product.

**Documentation:**

N/A

**Limitations:**

yes

**Opportunities For Improvement:**

The contribution that this paper makes with respect to prior work is not clear. Many of these issues are discussed in the related work the authors cite. Collecting those individual insights in one place (this paper) may be valuable, but the authors could do a better job distinguishing their paper from prior work and describing what is novel/unique/valuable about this paper that wouldn’t be gleaned from reading other papers on benchmark repos or benchmark practices.

**Relation To Prior Work:**

The authors do a great job weaving in prior related work. However, it is not clearly discussed how this work differs from previous contributions.

**Summary And Contributions:**

This paper discusses the landscape of benchmark repositories, i.e. repositories that store and curate datasets and tools to evaluate ML on them. It is mostly a perspective piece describing the desirable properties of benchmark repositories and some of the current issues in making them effective.

---

> ### Author Rebuttal · Authors · 2024-08-17
>
> Thank you for your thoughtful review and constructive feedback. We are pleased that you found our paper to address significant issues and be of interest to a broad ML audience. We respond to your critiques in detail below.
>
> __“The authors don’t explain clearly whether the recommendations they are making are a novel contribution in an of themselves. Many of the points they make and discuss are made elsewhere in literature, often in the papers introducing dataset repositories that they discuss. So it’s unclear how much of a novel contribution this paper provides.”__
>
> __“The contribution that this paper makes with respect to prior work is not clear… Collecting those individual insights in one place (this paper) may be valuable, but the authors could do a better job distinguishing their paper from prior work and describing what is novel/unique/valuable about this paper that wouldn’t be gleaned from reading other papers on benchmark repos or benchmark practices.”__
>
> Thank you for pointing this out—we agree that it is important to clarify the relationship between our paper and related work. While many of our recommendations and examples draw from existing work, our paper is novel in defining benchmark repositories, in discussing best practices specific to them (e.g., in terms of leaderboarding), and in making the connection between these repositories, data practices in ML, and best practices for data repositories in general. To express this we will add the following text to the end of the first section of the paper:
>
> _To the best of our knowledge, this paper is the first to define and establish best practices specifically for benchmark repositories, as well as to connect the practices of benchmark repositories to both ML and data repository practices in general._
>
> __“I would have liked to see more insight into how maintenance of benchmarking repositories could be made to align with academic incentives. The authors point out the issue that teams of people are needed to maintain benchmark repositories, and point to new venues of publication, but stop short of recommending any solutions to the problem that keeping benchmark resources up to date is not incentivized in the academic world that supports ML research - that is, once a paper on a resource is written, there is diminishing return in maintaining its product.”__
>
> This is an important point that we now address in the text we added to the Discussion (which we explain in more detail in our response to Reviewer axrk).

---

> > ### Comment · Reviewer_iNCk · 2024-08-29
> >
> > Thanks for your response, it all makes sense

---

### Official Review · Reviewer_ieFZ · 2024-07-25
**Addresses important issues but causes confusion for reader due to choice of terminology and conflation of recommendations**

**Rating:** 4
**Confidence:** 4
**Clarity:** Yes.

**Review:**

See below.

**Strengths:**

- The authors identify benchmark repository owners as an actor in the AI evaluation ecosystem that can contribute to solving some of the persistent issues within the community
- The authors identify ways repositories can contribute to highlighting datasets as valuable scholarly contributions
- The paper seems to be the first to provide recommendations provided for repository owners
- The authors put an emphasis on two critical issues in benchmark practices, uncertainty quantification and reproducibility, and how to potentially address such issues
- They provide an overview of some issues with current benchmarking practices (e.g., mismatch between what's evaluated vs. real-world task, introducing challenges for benchmark users)

**Additional Feedback:**

N/A

**Correctness:**

The claims seem to be supported by other literature, but no empirical evidence has been provided.

**Documentation:**

N/A

**Ethics:**

No.

**Limitations:**

- Right now, this paper lacks a discussion of its own limitations.
- The authors offer little to no quantitative analysis that backs up any of their claims surrounding best practices, issues, etc.; recommendations are based on literature review and authors' experience, but lack empirical validation
- The authors could explore in more depth the resource requirements and incentives needed to realize these improvements
- It is unclear how repository owners should conduct quality reviews, given that (as the authors note earlier) they are not necessarily the ones who have enough domain-knowledge about the benchmark to provide a meaningful evaluation. It seems like quality evaluations will largely be user driven on these platforms.

**Opportunities For Improvement:**

- The term 'benchmark repositories' is a confusing choice to the reader, as it's also used for benchmarks hosted by benchmark developers on, e.g., Github. I would suggest a name change.
- Furthermore, the authors introduce two terms for the same concept ('benchmark repositories' and 'benchmark dataset repositories'), which adds to the confusion.
- The paper dedicates significantly more space to describing issues than to describing solutions, which causes the descriptions of the solutions to be sometimes too high level to be actionable. For example, when describing issues around compositional and task metadata, the benefits and issues are discussed in 2.5 paragraphs, while the accompanying recommendation for repository owners is one sentence long. This is a recurring pattern for most recommendations.
- A more in-depth discussion on what compositional and contextual metadata should be included to enable benchmark users to make a decision on whether a benchmark is good would have been helpful.
- The authors should discuss limitations of their work.

**Relation To Prior Work:**

There has been no discussion on whether a similar provision of best practices for repository owners has been done in the literature. If the authors have checked this, it would be good if they can add something along the lines of "to the best of our knowledge, this paper is the first to provide best practices for repository owners".

**Summary And Contributions:**

This paper examines the role of what the authors call 'benchmark repositories' in ML research, focusing on how these group of actors can address criticisms of current data and benchmarking practices. The authors introduce the term 'benchmark repository' to describe repositories that support the discovery and use of datasets for evaluating ML models. They review issues throughout the dataset lifecycle and provide recommendations for how benchmark repositories can help address these problems.

---

> ### Author Rebuttal · Authors · 2024-08-17
>
> Thank you for your detailed review and constructive feedback. We appreciate that you found our paper to have the potential to address some persistent issues in the community, and we respond to your critiques in detail below.
>
> __“The term 'benchmark repositories' is a confusing choice to the reader, as it's also used for benchmarks hosted by benchmark developers on, e.g., Github. I would suggest a name change.”__
>
> __“Furthermore, the authors introduce two terms for the same concept ('benchmark repositories' and 'benchmark dataset repositories'), which adds to the confusion.”__
>
> Thank you for pointing out this potential avenue of confusion. We introduce the term “benchmark data repository” or “benchmark repository” for the type of data repositories that we discuss because existing terminology from the literature on data repositories (e.g., “generalist repository”, “domain-specific repository”, or “institutional repository” https://www.nlm.nih.gov/NIHbmic/generalist_repositories.html, https://zenodo.org/records/7051125, https://eric.ed.gov/?id=EJ1144764, https://doi.org/10.5334/dsj-2016-006) does not capture the uniqueness of data repositories created for ML benchmarking. For brevity throughout the paper, we primarily use “benchmark repository” and clarify in the introduction that this is interchangeable with “benchmark data repository” in the paper. There is some precedent for this kind of naming in the data repository community (e.g., in the sources shared above, the word “data” is often omitted). We can understand, however, that for the general ML audience, having the word “repository” without the qualifier of “data” preceding it can lead to confusion with other kinds of repositories, including those on GitHub. Minimally, we could change this sentence from the introduction
>
> |   “In this paper, we introduce the term _benchmark repository_ (or benchmark dataset repository) to describe repositories that support the discovery and use of datasets for evaluating ML models”
>
> to
>
> |   “In this paper, we introduce the term _benchmark data repository_ (or, for brevity, _benchmark repository_) to describe …”
>
> Instead, we are also open to using the term “benchmark data repository” throughout the paper and in the title (see also our response to reviewer kc5S) if the reviewers generally feel that this would improve the paper’s clarity.
>
> __“The paper dedicates significantly more space to describing issues than to describing solutions, which causes the descriptions of the solutions to be sometimes too high level to be actionable. For example, when describing issues around compositional and task metadata, the benefits and issues are discussed in 2.5 paragraphs, while the accompanying recommendation for repository owners is one sentence long. This is a recurring pattern for most recommendations.”__
>
> __“A more in-depth discussion on what compositional and contextual metadata should be included to enable benchmark users to make a decision on whether a benchmark is good would have been helpful.”__
>
> Because the goal of the paper is to present the ecosystem of benchmark repositories and how they connect to dataset/benchmarking issues in ML, we agree that we tend to spend more text in the paper on describing the landscape of data repository issues and their relationship with ML practices than we do on concrete recommendations. This is an intentional part of the paper’s structure; in each section, we describe the considerations in that area, provide relevant definitions, summarize benefits/issues, and discuss where benchmark repositories may play a role, providing supporting references that go into more detail in that particular area than we have space for in this paper. We do summarize numerous concrete suggestions in the paper (e.g., mint DOIs, require licenses), but the particular implementations of these suggestions are often repository-specific. For example, in section 4.1, we recommend that, to support data usability, benchmark repositories collect compositional and task metadata. What compositional and task metadata look like will differ for a repository that focuses on e.g., tabular data vs. text data. We certainly agree that detailed guidance for each of our recommendations would be highly useful, but it is beyond the scope of the paper to do so for every topic we discuss in this paper.
>
> __“The authors should discuss limitations of their work.”__
>
> __“The authors could explore in more depth the resource requirements and incentives needed to realize these improvements”__
>
> Thank you for pointing out that the limitations of our work were not made clear enough in the paper. We have added a “Limitations” subsection to the Discussion section in the paper that discusses the challenges of incentivization, which we provide more detail on in response to Reviewer axrk.

---

> > ### Author Rebuttal · Authors · 2024-08-17
> >
> > __“The authors offer little to no quantitative analysis that backs up any of their claims surrounding best practices, issues, etc.; recommendations are based on literature review and authors' experience, but lack empirical validation”__
> >
> > In general, we believe the claims we make about current issues are well supported by concrete results in the references we cite and specific examples throughout the paper. However, in light of your suggestion, we have edited several passages throughout the paper where we agree that additional references and/or explicitly stating the numerical findings from existing references could strengthen our argument (see details in our response to reviewer zDCB).
> > In terms of quantitative results about our recommendations: whenever possible, we point to empirical evidence about the effects of the practices we suggest. For example, our recommendation to associate datasets with a responsible point of contact is supported by a survey on reproducibility in machine learning research (see the end of Section 3.1 in [118]). For recommendations where these results are not available (e.g., because the practice has not yet been implemented at scale), we believe our justifications are largely intuitive and sufficiently supported by well-established standards and case studies. Although we agree that precisely quantifying the effects of each recommendation would be interesting future work, it is out of the scope of this paper.
> >
> > __“It is unclear how repository owners should conduct quality reviews, given that (as the authors note earlier) they are not necessarily the ones who have enough domain-knowledge about the benchmark to provide a meaningful evaluation. It seems like quality evaluations will largely be user driven on these platforms.”__
> >
> > We certainly agree that how to implement a review process for datasets at benchmark repositories is a challenging open question. From the case studies we present in the paper, it is clear that external user dataset audits are extremely valuable in identifying issues after the fact, but for pre-release reviews, it is indeed impractical for benchmark repository teams to have domain expertise in every field from which they might receive a dataset. To better address this point, we have added a discussion to section 3.2 in the paper about options for outsourcing this work. For example, benchmark repositories could partner with the Data Curation Network (DCN) https://datacurationnetwork.org based at the University of Minnesota. The DCN is a cross-institutional shared-staff network; if an institution does not have the expertise required to review a dataset, it can send it to the network for review. As we discuss in more detail in our response to Reviewer axrk, we have also added to our Discussion section that another potential useful model for this is “public journals”, such as JMLR, which use a volunteer-based reviewing system; another idea is for repositories to implement a similar system.
> >
> > __“There has been no discussion on whether a similar provision of best practices for repository owners has been done in the literature. If the authors have checked this, it would be good if they can add something along the lines of ‘to the best of our knowledge, this paper is the first to provide best practices for repository owners’.”__
> >
> > This is a good point; we have added text to this effect to the final version of our paper (see response to Reviewer iNCk for details).

---

> > > ### Comment · Reviewer_ieFZ · 2024-08-23
> > > **Rebuttal Response**
> > >
> > > Dear authors,
> > >
> > > Thank you for your detailed response.
> > >
> > > 1) On the first point, in my opinion, using benchmark data repositories would make it significantly clearer throughout the paper. As mentioned, the term “benchmark repository” is ambiguous, as it is already used by many in the community meaning the (Github) repository of one specific benchmark, which is not what you mean. I would at least change the title to “Benchmark Data Repositories for Better Benchmarking” because using the abbreviated version here doesn’t convey what the paper is about, given the ambiguity mentioned above (which you also acknowledged and which is why you introduced the term ‘benchmark data repository’ in the first place in the paper, if I understand your explanation correctly).
> > >
> > > 2) On the second point, I disagree with this notion, and think this is where the paper looses a lot of value if you just link the benchmark repository ecosystem to broader benchmarking issues. As a reader, I'd want to understand how benchmark data repositories should be structured and what measures their owners should take to increase usability and mitigate existing issues in the benchmark community. This understanding should be based on a detailed analysis of existing repositories—their strengths and weaknesses. While you mention within one to two sentences per issues on a very high level what benchmark data repository owners could do, this information isn't sufficiently actionable or contextualized. I agree that an understanding for the issues in the benchmarking community and how they may (on a high level) be addressed by benchmark data repositories is helpful, but driving change requires providing detailed enough information about potential solutions. To achieve better practices within the overall benchmark community, a detailed discussion of best practices within the benchmark data repository community would have been necessary in my opinion,  analyzing best practices of benchmark data repositories, in which contexts they’re good, etc.
> > >
> > > 3) On the third point, thanks for including limitations!

---

> > ### Author Response · Authors · 2024-08-30
> > **Response followup**
> >
> > Thank you for your reply to our rebuttal.
> >
> > 1. Given that Reviewer kc5S also has similar concerns about the terminology, we currently plan to change the title of the paper to “Benchmark Data Repositories for Better Benchmarking” and will make a final decision on the terminology throughout the paper once we have the final reviews and the AC’s metareview. Thank you for your input on this.
> > 2. With this paper, we aim to introduce benchmark data repositories as _entities_ in the ecosystem of ML benchmarking and provide a holistic overview of where they fit into this landscape, particularly with respect to the important criticisms that plague the benchmark community. To our knowledge, there is not yet a cohesive list of best practices for benchmark data repositories as the issues surrounding these repositories are scattered throughout various bodies of literature; our paper also broadly seeks to unite these. We respectfully disagree that an in-depth analysis of the current practices of benchmark data repositories is in the scope of this paper given these goals, but we nonetheless believe that such a paper would be a valuable followup contribution, as would a detailed list of prescriptive recommendations for benchmark data repositories.

---

### Official Review · Reviewer_zDCB · 2024-07-25
**A promising proposal addressing the current deficiency in benchmark and dataset efforts**

**Rating:** 8
**Confidence:** 3
**Clarity:** This paper is well written and all po…

**Review:**

This paper is a high-quality work that investigates the deficiencies in current dataset and benchmarking efforts in the machine learning (ML) community, and proposes a solution to address these issues. The paper is well-written, original, and offers a significant solution to the current pain point of benchmarking and dataset community.

Pros:
- The paper provides a comprehensive examination and analysis of the problems prevalent in current dataset and benchmarking efforts.
- The authors propose a promising solution and clearly define its role, which may serve as a valuable foundation and guideline for future research.
- The paper is well-written, with clear and concise language that facilitates easy understanding.

Cons:
- Some of the discussions regarding deficiencies in the current system lack concrete results or numerical data to support the claims.

**Strengths:**

- The paper provides a comprehensive examination and analysis of the problems prevalent in current dataset and benchmarking efforts.
- The authors propose a promising solution and clearly define its role, which may serve as a valuable foundation and guideline for future research.
- The paper is well-written, with clear and concise language that facilitates easy understanding.
- This paper has the potential to promote more ethical and socially responsible use of current datasets and benchmarks, and may encourage further research to address previously underemphasized aspects.

**Additional Feedback:**

N/A

**Correctness:**

Although some of the claims are lacking concrete results or numerical data to support, I believe the analysis were conducted in a sound way.

**Documentation:**

Not applicable here since it's not proposing a dataset or benchmark.

**Ethics:**

There are no ethical concerns.

**Limitations:**

While it is mentioned in the paper that a dedicated team will likely be responsible for managing this repository, a significant amount of work might still fall on the dataset owners to input relevant information, such as compositional and task metadata. As a result, there is a risk that such data may be inaccurate, incomplete or worse manipulated, which could impact the effectiveness of the system.

**Opportunities For Improvement:**

- Some of the discussions regarding deficiencies in the current system lack concrete results or numerical data to support the claims.
- While this proposal shows promise, it remains unclear how the proposed system will be implemented in practice and how a sustainable ecosystem can be established and maintained.

**Relation To Prior Work:**

Relation to prior work is clearly discussed across all sections.

**Summary And Contributions:**

This paper introduced the concept of 'benchmark repositories', examined in detail their role in addressing criticisms of current data and benchmarking practices in ML research, and discussed various considerations for designing and using such 'benchmark repositories'.

---

> ### Author Rebuttal · Authors · 2024-08-17
>
> Thank you for your thoughtful review and constructive feedback. We are very pleased that you found our paper to be of high quality and significant impact to the field, laying the foundation for future research in this important area. We respond to your points in detail below.
>
> __“Some of the discussions regarding deficiencies in the current system lack concrete results or numerical data to support the claims”__
>
> In general, we believe the claims we make are well supported by concrete results in the references we cite and specific examples throughout the paper. However, in light of your suggestion, we have identified several passages throughout the paper where we agree that additional references and/or explicitly stating the numerical findings from existing references could strengthen our argument. In particular:
> * 2.3—current ambiguity about licensing in ML: we have added references to two surveys of current licensing practices—https://arxiv.org/pdf/2310.16787 and https://dl.acm.org/doi/abs/10.1145/3589334.3645520—and have added the following text to the paper: “current licensing practices for ML datasets are often irregular, conflicting, over-permissive, or poorly documented—in fact, in one survey of over 1800 text datasets, 69% had “unspecified” licenses on HuggingFace”
> * 3.1—under-documentation of data practices: we now elaborate on “documentation is often scattered and unstandardized,” pointing to a finding from [32]: “for example, in a survey of papers introducing computer vision datasets, of those that used human annotation, only 63.4% provided any description of the annotators and only 7.8% reported annotator demographics [32]”
> * 3.2—lack of ethical review: we now elaborate on “benchmark data collection often does not undergo institutional review board (IRB) or other ethical review,” pointing to another finding from [32]: “in fact, in [32], only 5% of papers introducing datasets with human subjects mentioned an IRB or equivalent ethical review”
> * 3.3—lack of standardized deprecation protocol: we now elaborate on “creators often withdraw their dataset without an explanation of why it was withdrawn or explicit instructions not to use the dataset,” pointing to a finding from [70]: “for example, in [70]’s case study of six high-profile dataset retractions, three (MS-Celeb-1M, Duke MTMC, and HRT Transgender) provided no explanation of why their dataset was removed”
> * 5.0—lack of statistical significance: we now elaborate on “seemingly record-breaking performance improvements are not always statistically significant,” pointing to a finding from [158]: “for example, a review of the MS MARCO leaderboard found that there was no significant difference between the current top three models”
>
> Beyond the findings from the papers we cite, we agree that a comprehensive quantitative characterization of current data practices in ML and in benchmark repositories would be valuable and interesting. However, we believe this is beyond the scope of the current paper (and could potentially distract from our high-level qualitative message).
>
> __“While this proposal shows promise, it remains unclear how the proposed system will be implemented in practice and how a sustainable ecosystem can be established and maintained”__
>
> This is an important point that we now address in the text we added to the Discussion (which we explain in more detail in our response to Reviewer axrk).
>
> __“While it is mentioned in the paper that a dedicated team will likely be responsible for managing this repository, a significant amount of work might still fall on the dataset owners to input relevant information, such as compositional and task metadata. As a result, there is a risk that such data may be inaccurate, incomplete or worse manipulated, which could impact the effectiveness of the system.”__
>
> We completely agree—metadata may be inaccurate, incomplete, or provided in bad faith. We do not view this as a limitation of the suggestions we make in the paper, however, but rather find that it helps support them. We believe that relying on data donors to provide accurate and holistic metadata is an even bigger risk _without_ having a repository in the loop since the onus is then completely on the data providers to 1) decide _what_ metadata to provide, and then 2) document it accurately. A structured metadata schema and quality review process, enforced by a repository, reduce this risk, particularly at step 1. This is a useful and important point that we have added to the end of Section 4.1 in the paper; thank you for raising it.

---

> > ### Comment · Reviewer_zDCB · 2024-08-30
> >
> > Dear authors,
> >
> > Thank you for providing the detailed response and addressing the feedback. I believe that this work will be highly insightful for the dataset and benchmarking efforts within the machine learning community.

---

### Official Review · Reviewer_kc5S · 2024-07-28
**The right way to write an ecosystem paper.**

**Rating:** 9
**Confidence:** 5
**Correctness:** Yes.
**Clarity:** Yes.

**Review:**

Significance/Impact (high): Benchmark repositories are hugely impactful tools for the AI field. This paper is very intelligently constructed as a review of all repos and analysis of how they could be improved. This is much more useful than "yet another dataset repo, and the few things we do differently from other repos" class of all-to-frequent paper.

Originality/Innovation (med): Good papers analyzing ecosystem pillars and presenting useful ideas on how to improve them are rare.

Clarity (med-high): the paper is well written and easy to follow.

Ethical/social (moderate-high): ecosystem contributions have wide benefits for everyone doing work in the field.

**Strengths:**

AI depends on datasets and benchmarks to enable and measure progress. The portals are vital resources.
Almost everyone submitting to neurips would be better off if this paper's recommendations were followed widely.

**Additional Feedback:**

Great paper on an important topic. This is not a showy paper, but one that is needed.

**Documentation:**

Yes.

**Ethics:**

No.

**Limitations:**

See areas of improvement.

**Opportunities For Improvement:**

Please say "Benchmark and Dataset" in the title -- this paper is at least as much about dataset repos in general.

**Relation To Prior Work:**

Yes.

**Summary And Contributions:**

The paper describes the need for dataset/benchmark repos. It talks about how it's hard to properly reward dataset creation, and issues that arise over the lifecycle of a dataset. The paper suggests ways to make data more reusable and benchmarks more reproducible. It also suggest ways to make benchmarks leaderboard drive improvements in testing practices, and encourage ongoing development of datasets.

---

> ### Author Rebuttal · Authors · 2024-08-17
>
> Thank you for your thoughtful review and kind words about our paper. We are very pleased that you found our paper to be a necessary and high-impact contribution that would benefit the field. We respond to your feedback on the paper’s title below.
>
> __“Please say ‘Benchmark and Dataset’ in the title -- this paper is at least as much about dataset repos in general.”__
>
> We appreciate your point that including “dataset” in our title could improve its clarity, and thank you for the suggestion. We are a bit hesitant that the suggested revised title, “Benchmark and Dataset Repositories for Better Benchmarking,” might not align with our intended specific focus on the ecosystem of benchmark repositories, although, as you point out, several of our points do apply to data repositories in general. Another option is to change the title to “Benchmark _Dataset_ Repositories for Better Benchmarking” or, similarly, to “Benchmark _Data_ Repositories for Better Benchmarking” to make explicit that our paper is about data repositories (which also addresses the point made by Reviewer ieFZ). Throughout the paper, we do use the term “benchmark repository” (and not “benchmark dataset repository”) for brevity, but we try to guard against this terminology confusion when we introduce it in the introduction. We are open to considering these alternative titles if it is generally agreed that an updated title will improve the clarity of the paper.

---

### Author Rebuttal · Authors · 2024-08-17

We would like to thank the reviewers for their time and valuable feedback. We are glad that reviewers generally found the paper to be an important, potentially high-impact contribution to the ML community (e.g., “wide benefits for everyone doing work in the field”, ”potential to promote more ethical and socially responsible use of current datasets and benchmarks”, “surfaces an important yet underdiscussed issue”). We respond to reviewer-specific concerns and suggestions in our individual rebuttals.

Several reviewers noted that our paper provides an important foundation for future discussion and research into data repository issues in ML, but that the paper would be strengthened by a more detailed discussion of the challenges and uncertainty involved in incentivizing the implementation and maintenance of our suggestions. We agree and now elaborate on these challenges in a “Limitations” subsection in the Discussion section of our paper using the additional allotted space. In this section, we acknowledge that the long-term feasibility and impact of our suggestions are predicated upon larger shifts in community norms and attitudes about data-centric work, which will rely upon proper incentivization, an important open challenge in the ML community (e.g., refs 24, 27, 28). We go on to discuss potentially useful avenues of incentivization from which repositories may be able to draw inspiration: research on scholarly ecosystems for sharing research data (e.g., https://doi.org/10.1002/asi.22634, https://doi.org/10.5334/dsj-2017-014, https://mitpress.mit.edu/9780262529914/big-data-little-data-no-data/), volunteer-based journals such as JMLR, and increased attention from funding agencies and institutions on data-based services (e.g., ref 130 and https://grants.nih.gov/grants/guide/notice-files/NOT-OD-21-013.html). We respond in more detail on these issues in our reply to Reviewer axrk. We are glad that the reviewers generally expressed that our paper provides an important first step in initiating discussions and research on these kinds of issues for benchmark repositories.

---

### Decision · Program_Chairs · 2024-09-26

**Decision:**

Accept (Poster)

**Comment:**

While this paper has a high average rating (7.0 after an active discussion), there seems to be a clear story connected to it that needs to be dived deeper into before a weighted decision of acceptance can be made.

* reviewer kc5S is positive about the paper, but remains high-level in the feedback, citing that the community is better off when recommendations from papers like these are followed. I would agree with the reviewer that statements like this are important, but without a qualitative assessment of why the quality of this paper is so high, it might not entirely warrant this specific grade.
* reviewer zDCB is also positive, but mentions "some of the discussions regarding deficiencies in the current system lack concrete results or numerical data to support the claims". This is resolved, and the reviewer has acknowledged this.
* reviewer ieFZ mentions several points for improvement, to which the most important seem to be "the authors should discuss limitations of their work" (seems resolved), and "The paper dedicates significantly more space to describing issues than to describing solutions, which causes the descriptions of the solutions to be sometimes too high level to be actionable" (while rebutted, this is not resolved).
* reviewer iNCk raises a concern about which parts are novel, and which ideas were floating around in literature. Eventually, the authors seem to have resolved this.
* reviewer axrk is mostly positive, but also raises a point about the actionability.

All together, it seems that the community has responded mostly advocating accept (3), with 1 neutral and 1 more critical review. While I see the points for improvement that are raised in both the positive and negative reviews, and I do think these are important, I feel that the peer-review has largely decided that this paper should be accepted. However, I would suggest the authors to take serious consideration of the more critical suggestions made in this review, and consider them either for the CRC version (in case these are minor) or for future work.